# Kinetic Investigations on the Chiral Induction by Amino Acids in Porphyrin J-Aggregates

**DOI:** 10.3390/ijms24021695

**Published:** 2023-01-15

**Authors:** Roberto Zagami, Maria Angela Castriciano, Andrea Romeo, Luigi Monsù Scolaro

**Affiliations:** 1Dipartimento di Scienze Chimiche, Biologiche, Farmaceutiche ed Ambientali, University of Messina, V.le F. Stagno D’Alcontres, 31-98166 Messina, Italy; 2CNR-ISMN Istituto per lo Studio dei Materiali Nanostrutturati c/o Dipartimento di Scienze Chimiche, Biologiche, Farmaceutiche ed Ambientali, University of Messina, V.le F. Stagno D’Alcontres, 31-98166 Messina, Italy

**Keywords:** porphyrins, J-aggregates, aggregation kinetics, amino acids, symmetry breaking, chiral supramolecular assemblies

## Abstract

The self-assembling kinetics of the 5,10,15,20-*tetrakis*(4-sulfonato-phenyl)porphyrin (TPPS_4_) into nano-tubular J-aggregates under strong acidic condition and in the presence of amino acids as templating chiral reagents have been investigated through UV/Vis spectroscopy. The ability of the chiral species to transfer its chiral information to the final J-aggregate has been measured through circular dichroism (CD) spectroscopy and compared to the spontaneous symmetry breaking process usually observed in these nano-aggregates. Under the experimental conditions here selected, including mixing protocol, we have observed a large difference in the observed aggregation rates for the various amino acids, those with a positively charged side group being the most effective. On the contrary, these species are less efficient in transferring their chirality, exhibiting a quite low or modest enhancement in the observed dissymmetry g-factors. On the other side, hydrophobic and some hydrophilic amino acids are revealed to be very active in inducing chirality with a discrete increase of intensity of the detected CD bands with respect to the spontaneous symmetry breaking.

## 1. Introduction

J-aggregates of dyes are subject to extensive research due to their peculiar structural and photophysical properties [1]. The electronic coupling between adjacent chromophores and the mutual geometrical arrangement determines important changes in the corresponding electronic spectra. According to the exciton model, an edge-to-edge interaction of aligned electronic transition moments in a dimer leads to an lower allowed energy level, which is revealed as a bathochromic shift of the corresponding band (J-band) [2]. Aggregation in porphyrins is a well-documented phenomenon [3,4], and formation of J-aggregates in aqueous solutions has been early reported [5,6] and then increasingly studied [7,8,9,10,11,12]. In this framework, the tetra-anionic 5,10,15,20-*tetrakis*(4-sulfonatophenyl)porphyrin (TPPS_4_) has received greater attention since it is fairly soluble in water and its aggregation can be easily initiated by a variety of experimental conditions. Lowering pH, this compound is easily protonated at the core first [13] and then at the peripheral sulfonate groups [14]. Dianion and zwitterion forms are able to self-assemble into a variety of aggregates, whose size and morphology strictly depend on concentration, pH, temperature, ionic strength and mixing protocols [14,15,16,17,18,19,20,21,22,23]. Among the various possibilities, nanotubes of porphyrins form when strong acidic conditions are used [24]. The resulting J-aggregates are promoted and stabilized by an interplay of intermolecular forces, including electrostatic, H-bonding and solvophobic interactions [20,21,25,26,27,28,29,30,31,32,33,34]. In this context, the addition of a series of templating species enlarges the number of examples and expand the complexity of these nanoaggregates [25,26,27,35,36,37,38].

A fascinating property emerging in some of the reports on these species is the observation of circular dichroism bands in the visible region of their spectra, implying they are chiral. In these aggregates, chirality can be either induced through chemical [15,25,35,39,40,41,42,43,44,45,46,47] or physical bias [48,49,50,51,52,53,54,55], or originating from spontaneous symmetry breaking [20,22,23,56].

Among the plethora of chiral compounds proposed in the literature to induce chirality into supramolecular porphyrin systems, amino acids are largely used. Apart from porphyrins chemically modified to incorporate amino acid residues [57,58,59,60], examples of non-covalent approaches, i.e., where chirality is fostered onto self-assembled arrays of achiral porphyrins through simple amino acids added in solution, are less common [61,62,63,64,65,66]. Quite recently, Randazzo et al. showed that chirality transfer from specific amino acids to porphyrin J-aggregates is a process under hierarchical control [47]. Indeed, the mixing protocol revealed to be decisive to effectively imprint the handedness of the chiral bias to the growing supramolecular structure. In our previous investigations on the expression of chirality in TPPS_4_ J-aggregates, we pointed out the paramount importance of the reagents mixing sequence. It determines the kinetic growth and consequently the eventual detection of CD bands for the aggregated species, either in spontaneous symmetry breaking (i.e., in the absence of added chiral inducer) [23] or when a chiral template is deliberately added to the solution [14]. In addition, we observed an interesting case of kinetic discrimination when the two enantiomers of tartaric acid were used to induce chirality in these nano-aggregates [41]. Due to the complexity of the aggregation pathway, the encounter between the porphyrin building block and the chiral inducer is crucial, especially if expression of chirality is the final goal [14]. Furthermore, both the TPPS_4_ porphyrin [67] and the amino acids exist under different forms with different overall charge depending on pH. Since the chiral transfer and the self-assembling processes are driven by a combination of hydrophobic and solvophobic interactions, the exact knowledge of the nature of the species and their contact-timing along the aggregation process is fundamental.

Here we report on the impact that various amino acids exhibit on both the rates of porphyrin self-assembling process and the intensity and sign of the ICD signals observed for such aggregated systems. The amino acids have been chosen in order to have hydrophobic (isoleucine (Ile), proline (Pro), valine (Val)) or hydrophilic (serine (Ser), threonine (Thr), citrulline (Cit)) side chains, with positively (arginine (Arg), lysine (Lys), ornithine (Orn)) or negatively (aspartic acid (Asp)) charged groups, with a wide spread in their isoelectric points (IEP) (Appendix A). Under the conditions selected for these experiments, we anticipate that specific kinetic effects are detected for various classes of such compounds, being those bearing a positively charged side chain the less effective in transferring their chirality to the growing supramolecular assemblies.

## 2. Results and Discussion

TPPS_4_ porphyrin exists under various forms with different charges and protonation degree as function of pH (Figure 1). At neutral pH, the four sulphonato groups are ioni0zed and consequently the porphyrin is a tetra-anion (TPPS_4_^4−^), characterized by an intense B-band at 414 nm. At pH ranging from 5 to 3 the main prevailing species is a di-anion (H_2_TPPS_4_^2−^), due to the protonation of the two central nitrogen atoms [13]. This species features an intense red-shifted band at 434 nm. Below pH 3, further protonation of two sulphonato groups leads to a neutral zwitterion (H_4_TPPS_4_), with no clearly distinguishable spectral changes from the dianion beside a quite modest hypochromic effect. As function of pH, ionic strength and/or other templating reagents, these two latter species can eventually self-organize into J-aggregates, whose spectroscopic markers are a narrow and intense peak at 490 nm (J-band), along with a minor feature at 422 nm (H-band) [1,5].

In our experience when an aqueous solution of an inorganic acid is added to a diluted TPPS_4_ solution, aggregation occurs following an autocatalytic path with a well-defined incubation period. This experimental approach has been named *porphyrin first* protocol (PF) and eventually leads to spontaneous symmetry breaking and detection of CD signals for the final aggregated species [23]. As pointed out in the previous section, the addition of a chiral templating reagent requires some attention due to potential ageing effects, thus affecting the eventual results. Therefore, we decided to trigger the aggregation using a modified PF protocol, in which a premixed solution of inorganic acid (H_2_SO_4_) and the selected amino acid is added to a prediluted porphyrin solution in a 1:1 (v/v) mixing protocol (see Materials and Methods). The actual concentration of all the species in the reacting mixture ([TPPS_4_] = 10 μM, [H_2_SO_4_] = 0.5 M, [amino acid] = 0.1 M) have been selected to ensure that the aggregation kinetic parameters can be easily determined from changes in their UV/Vis extinction spectra. The concentration of porphyrin has been chosen to lead to a minimal extent of spontaneous symmetry breaking [23,68]. The pH of the solutions ensures also that we deal with the neutral zwitter-ionic, H_4_TPPS_4_, porphyrin that should self-assemble into nanotubes. Considering the values of the IEP for all the investigated amino acids, Arg, Lys and Orn have all their amino groups in a protonated form, therefore they are dicationic species. The other amino acids are in the form of mono-cations.

Figure 1A shows the typical spectral changes observed for the aggregation of the porphyrin in the presence of L-serine. Indeed, formation of the J-aggregates is evident from the increase of the extinction at 492 nm, paralleled by a matching decrease of the extinction at 434 nm (inset Figure 1A). Furthermore, a distinct and quite flat lag-time is well-evidenced in the early stages of the kinetic traces.

The CD spectra of the final samples reveal signals induced in the region of the J- and H-components for the J-aggregates (Figure 1B), with the typical exciton-split couplets that are positive for L- and negative for the tested D-enantiomer (Ser, Val, Asp). All these spectral features are common for the amino acids bearing a single positive charge (Ile, Pro, Ser, Thr, Val, Asp, Cit).

In the case of dicationic amino acids (Lys, Arg and Orn), the UV/Vis spectra display the formation of a broader and less intense J-band, often accompanied by a shoulder at a longer wavelength, and with non-zero extinction all over the observed spectral range. Figure 2A reports a typical UV/Vis spectral change for the aggregation of TPPS_4_ in the presence of L-Arg, as a representative example. The time evolution of the extinction at 492 nm (or 434 nm) shows a much shorter and steep lag-time with respect to the other amino acids (inset Figure 2A). This spectroscopic evidence is not unprecedented, since it was reported in the case of polyamine induced TPPS_4_ J-aggregates [21,28,69]. The unusual J-band originates from a dipolar coupling mechanism among nanoaggregates and free monomeric porphyrins linked by bridging electrostatic interactions involving the positively charged polyamine [20,70].

The CD spectra recorded on the final aggregated samples evidence weak induced bisegnated CD bands in the J- and H-region. It is worth to note that (i) like the other amino acids, the sign of the observed exciton coupled band of the J-component is related to the specific handedness of the templating amino acid (positive for D-Arg and negative for L-Arg and L-Lys); (ii) for Arg the sign of the CD band in the J- and H-bands are mutually inverted; (iii) the relation between sign of the exciton CD J-couplet and handedness of the amino acid is inverted with respect to the mono-cationic amino acids. This latter observation has been already reported for L-Arg as compared to L-Lys, L-His and L-Phe by Randazzo et al. [47]. Anyway, it is worthwhile noting that the sign of the induced CD bands detected in our work are opposite in sign (we observed positive exciton splitting for all the L-amino acids except for L-Arg and L-Lys) and much lower in intensity. A possible explanation for this apparent discrepancy could be based on the different structural arrangements in J-aggregates that have been prepared under different experimental conditions and mixing protocols. Indeed, previous investigations suggest that when aggregation of TPPS_4_ is fostered in the presence of tartaric acid, the sign of the observed CD bands for the same enantiomer switches as a function of the different mesoscopic structure, which is also controlled by ionic strength and pH [18].

Concerning the kinetic analysis of the extinction data, we exploited a well-established model proposed by Pasternack et al. [71,72]. It assumes that the formation of an initial nucleus containing *m* porphyrin units is the rate-determining step that leads to an autocatalytic growth. This pathway is governed by a power law with a typical time-exponent *n* and a rate constant *k_c_*. The model takes also into account a potential non-catalyzed pathway with a rate constant *k_0_*. All the relevant kinetic parameters determined by non-linear best fitting of the extinction data at 492 nm are collected in Table 1, together with the calculated dissymmetry *g*-factors at the J-bands. For the sake of comparison, the relevant data for aggregation of TPPS_4_ triggered by H_2_SO_4_ using a PF protocol are also shown.

Figure 3 displays a comparison of the values for the autocatalytic rate constant *k_c_* in the presence of the various amino acids, both as absolute values (Figure 3A) and their percent deviations (Figure 3B) from the reference (rate for the aggregation induced by H_2_SO_4_ only, dashed line). From inspection of both graphs, it is evident that a substantial number of amino acids provokes only a slight acceleration or deceleration with respect to the reference (within 20%). D-Asp, D-Arg, L-Arg and L-Lys are effective in promoting a more relevant rise of the aggregation rate, and in particular, both enantiomers of Arg exhibit a more than two-fold increase of the corresponding values.

The values of the rate constant for the not-catalyzed pathway, *k_0_*, follow a trend similar to the *k_c_* parameter (Figure 4). In this case, the increase of the rates is much more consistent, being almost 20-fold in the case of L-/D-Arg and 5-fold for L-Lys and D-Asp. This experimental finding is in line with the higher slope in the initial lag-time of the kinetic traces for these systems (inset Figure 2A versus inset Figure 1A).

The number of monomer units in the initial nucleus, *m*, in spontaneous symmetry breaking induced by inorganic acids ranges between 3–4, suggesting that the formation of a trimer or a tetramer is involved in the rate determining step of the aggregation process [23,68]. In the presence of amino acids, the observed values are in the same range, with the exception for L-/D-Arg, L-Lys, L-Orn and L-Cit (Table 1, Appendix A). It is noteworthy that, with the exception of the last one, all these amino acids are dicationic species in the acid conditions of our experiments. It is interesting to observe that also the values for the power exponent *n*, that is indicative of the growth process, are generally close to the value observed for the reference system (H_2_SO_4_, *n* = 7.9), with the exception of the dicationic species L-/D-Arg, L-Lys, L-Orn (*n* in the range 1.7–3.0) (Table 1, Appendix A). The low values for both *n* and *m* indicate a changeover in the aggregation mechanism when passing from the dicationic to the monocationic amino acids. This fact is further supported by the broadening of the J-band in the UV/Vis spectra of the corresponding aggregated samples (Figure 2A), that suggests the formation of mesoscopic structures and a different mechanism of electronic coupling among porphyrins in the aggregated species, as discussed above [28,69]. As a consequence of the broadening, the values of the extinction after completion of the aggregation process in the case of the dicationic species is largely reduced in comparison to the reference and the monocationic species (Appendix A). In order to compare the difference in the effective transfer of chirality from the amino acid to the J-aggregates, we exploited the dissymmetry *g*-factor, that is related to the quality of the CD rotational oscillator and is uncoupled with respect to concentration of the specific sample (*g* = Δε/ε = ΔA/A, where Δε = ε_L_ – ε_R_ and ΔA = A_L_ - A_R_) [73]. Under the experimental conditions used in the present study, when aggregation is triggered by adding H_2_SO_4_, we detect small induced CD bands at the J- and H-bands, indicative of a spontaneous symmetry breaking process that leads to a scalemic mixture of chiral nanotubes with a slight enantiomorphous excess toward the P-helix (g = +0.47). The extent of this excess and the intensity of the observed *g*-factor are strongly related to the aggregation rates *k_c_*. On accelerating the aggregation process, the *g*-values undergo to a substantial decrease [22,23]. Figure 5A reports the absolute values of the *g*-values collected in Table 1 for the various amino acids and the reference sample. It is worthwhile to remember that for all the L-enantiomer (except for L-Arg and L-Lys) positive values have been measured. An inspection of both Figure 5A,B reveals that the dicationic compounds Arg and Lys show values very close to the reference sample, while Orn has almost a three-fold increment of the *g*-factor.

If we exclude citrulline, that has a g-value very close to that of ornithine, all the other amino acids exhibit a large enhancement (close to 10-fold) in comparison to the reference, reaching 28-fold and 25-fold for L-Thr and D-Val, respectively. Figure 6 shows a comparison of the actual g-values (orange bars) and corrected ones (green bars), for pairs of enantiomers of selected amino acids. The corrected values have been obtained by subtracting the value of the reference. In both cases, it is evident that the enantiomers transfer their handedness to the supramolecular aggregates with similar efficiency. Even in the case of Asp, where the aggregation rates display the largest difference, the induced CD bands have a mirror relationship. The actual results are dissimilar to those already reported for the case of the tartaric acid in the aggregation and chiral induction with TPPS_4_ porphyrin, where kinetic and chiral discrimination have been observed for the enantiomeric pair [41]. Once again, it is important to stress that slight changes in the mixing protocol could be the cause of the observed different behavior.

In previous investigations, we have already pointed out that a scaling law holds when the dissymmetry *g*-factor is correlated to the aggregation rate *k*_c_ in the case of a spontaneous symmetry breaking process [22]. Figure 7 shows a plot of these two parameters for all the amino acids tested in our investigation (including the reference). Our experimental data seem not to follow a clear power law, even if two data sets could be identified (marked in the dashed boundaries lines). The set enclosed in the orange line includes all the dicationic amino acids, except than L-citrulline. All these species have aggregation rates that spread from slightly lower (L-Orn) than the reference to the largest one (D-Arg). Moreover, they are scarcely effective in transferring their chirality to the growing porphyrin aggregates. The reasons could not be related to the aggregation rates, but to a different kind of interaction between the porphyrin building block and the amino acids. Due to the double positive charge on this latter and the zwitterionic character of the porphyrin, the intermolecular interaction should be more electrostatic and resembling that already described for the aggregation of TPPS_4_ porphyrin with polyamines, such as spermine [21,28,69]. Our spectroscopic evidence (very large J-bands in the UV/Vis spectra) support this hypothesis. It is interesting to outline the difference between our findings and those reported by Randazzo et al. [47], which could find an explanation in the different mixing protocol, besides the pH and ionic strength conditions used in the experiments. Considering the second set enclosed in the blue dashed line, it gathers almost all the monocationic species (with the exception of L-Cit). If we exclude D-Asp, these amino acids have very similar values of the autocatalytic rate constants *k_c_*. J-aggregates of TPPS_4_ grown in the presence of L-Thr display the largest value of g-factor, with D- and L-Val only slightly less effective. These two amino acids are structurally very similar, differing only for the presence of a hydroxyl group (Thr) versus a methyl group (Val) on the β-carbon atom. If the role of the H-bonding donor group (OH) could be envisaged, serine and aspartic acid seem effective to the same degree or even less than the more hydrophobic L-Ile and L-Pro. Therefore, all the experimental evidence suggests that an interplay of interactions and probably size effects should be operative in determining the chiral transfer efficiency. In previous kinetic investigations, we have pointed out the role of “added” components, i.e., cationic metal complexes [20,22,74], cationic porphyrins [19] or simple anions [14,67] on the self-assembling mechanism of TPPS_4_. The early stage of the aggregation process is the formation of an initial nucleus containing *m* porphyrin units. This seed is able to collect and self-organize porphyrin monomers to the eventual nano-aggregate in an autocatalytic pathway. Anyway, the nucleation phase is the rate-determining step of the process and modulation of the rates are governed by the interaction among reacting species in this specific stage. When dicationic amino acids are used, a dimer is most probably involved in this stage with respect to monocations or to the spontaneous symmetry breaking in the absence of chiral inducers, where a trimer or a tetramer is the initial seed. Also, the change in the power exponent *n* is rather meaningful since it is related to the formation of a new reactive surface in the growing aggregate.

## 3. Materials and Methods

### 3.1. Materials

The sodium salt of 5,10,15,20-*tetrakis(*4-sulfonatophenyl)porphyrin (TPPS_4_) was purchased from Aldrich (Milan, Italy). All the amino acids (L-arginine, D-arginine, L-serine, D-serine, L-valine, D-valine, L-aspartic acid, D-aspartic acid, L-Isoleucine, L-threonine, L-proline, L-citrulline, L-ornithine, L-lysine) and H_2_SO_4_ were of the highest commercial grade available and were used as received without further purification from Sigma-Aldrich (Milan, Italy). All the aqueous solutions were prepared in high-purity doubly distilled water (HPLC grade, Fluka). Precautions were taken to avoid contamination during the preparation of the samples. A stock solution of porphyrin (100–200 μM) was freshly prepared and stored in the dark to avoid photo-degradation. The concentration used in the experiments was calculated by UV/Vis absorption spectroscopy using the molar extinction coefficients at the B-band (TPPS_4_: 5.33 × 10^5^ M^−1^cm^−1^, λ = 414 nm).

### 3.2. Methods

UV/Vis extinction spectra were collected on an Agilent 8453 diode array spectrophotometer. Photodamage on the porphyrin solutions during the kinetic runs was prevented by using an UV filter (Hoya glass type UV-34, cut-off: 340 nm) set between the lamp and the samples. Kinetic experiments were performed by recording full spectral changes on quartz Hellma cells placed in the thermostatic holder of the spectrophotometer. The mixing protocol used in our experiments is a modification of the previously reported *porphyrin first* (PF) [23]. 0.5 mL of a premixed aqueous solution of H_2_SO_4_ 1 M and the selected amino acid 0.2 M is added to 0.5 mL of a prediluted TPPS_4_ 20 μM solution in a 0.5 cm path length cell. The aggregation process is starting by inverting three times the cell. All the solutions and the samples have been kept at 298 K.

The kinetic analysis of extinction data (492 nm) versus time have been performed by a non-linear least square fit to the equation: Ext_t_ = Ext_∞_ + (Ext_0_ − Ext_∞_) (1 + (*m* − 1) {*k_0_*t + (*n* + 1)^−1^ (*k_c_* t)*^n^*^+1^})^−1/(*m*−1)^
(1)
where Ext_0_, Ext_∞_, *k_0_*, *k_c_*, *m* and *n* are the parameters to be optimized [72].

CD spectra were recorded on a Jasco J-710 spectropolarimeter. The values of the dissymmetry *g*-factor (*g* = Δε/ε = ΔA/A) were obtained from CD and UV/Vis extinction data by dividing ΔA (calculated from the total amplitude of the negative and positive component of the bisegnated J-band at 492 nm) by the extinction at the J-band.

## 4. Concluding Remarks

J-aggregates of TPPS_4_ exhibit hierarchical structures whose complexity is mainly controlled by the kinetic of growth, mixing protocols and a variety of added templating reagents. Under strong acidic conditions nanotubes are usually formed that exhibit an inherent chirality due to a helicoidal internal structure [24]. The mechanism that has been proposed is based on the rate determining formation of an *m*-mer of porphyrins able to auto-catalyze the subsequent grow of the final structure. Therefore, since the amino acids differ in their propensity to increase or decrease the aggregation rates, their involvement is suggested in the formation of the initial nuclei. We have shown that some amino acids are more effective in influencing the spontaneous symmetry breaking occurring in these systems, leading to an enhancement of the induced CD signals. It is clear that the knowledge of the protonation state of both porphyrin and chiral inducer is required in order to establish the nature of the interactions involved in the self-assembling process and in the chirality transfer mechanism. The exact role of each amino acid along the aggregation pathway is still elusive since a complex interplay of factors operates among the components of these supramolecular buildings.

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
