# Peer review of "Kinetic Investigations on the Chiral Induction by Amino Acids in Porphyrin J-Aggregates"

_ijms, 2023, doi:10.3390/ijms24021695_

Round 1

Reviewer 1 Report

In the present article, Zagami and coworkers realize a comprehensive study on the self-assembling kinetics of TPPS4 leading to J-aggregates in the presence of 10 amino acids, 4 of them used in their L and D isomeric form.

The study clearly indicates that some amino acids are more effective in influencing the spontaneous symmetry breaking of H4TPPS4 which leads to the formation of tubular supramolecular structures. As a result, a significant enhancement of the induced CD signal is observed.

It is interesting to point out that a similar study using TPPS and four (D and L) amino acids (two of them also used in the present study) was previously reported by another research group (ref. 48). In that case, different results were reached in terms of the capability of chirality transfer of the two amino acids also used in this study (i.e., Arg and Lys). The authors of the present work convincingly justify such different results on the basis of the different mixing protocol used, a determining factor, as important as others such as the pH and ionic strength.

I reckon that this manuscript offers a valuable insight into a stimulating issue as it is the influence of chiral species, in this case amino acids, in the kinetics of the self-assembly of a well-studied chromophore such as TPPS. Although, the authors tried to rationalize the results obtained tacking into account parameters such as the hydrophobic or hydrophilic nature of the amino acids used or the presence of a positive charge in the side chain, as stated by the authors themselves, the exact role of each amino acid in the aggregation pathway of TPPS is still to be fully understood since a complex interplay of factors operates.

The paper is well written, the figures are clear and well-presented, and the references appropriate so that I recommend its publication in International Journal of Molecular Science after addressing the comments/realizing the corrections reported below.

The references section needs to be thoroughly revised.

In some cases, the DOI is reported, in other cases not.

In some cases, the name of the journal is not included (e.g., references 60, 61, 62, 67, etc.).

Some references are duplicated (I spotted references 54 and 55, and references 30 and 40).

In some cases, the journal names are abbreviated, in many other cases not (e.g., references 72, 76, 71, 63, etc.).  

In some cases, the titles appear with only the first letter capitalized, in other cases the first letter of every name is capitalized.

The authors are invited to revise all the references and change them accordingly following the International Journal of Molecular Science references guidelines.

In Table SI1, the molecular structure of the amino acids with D and L handedness (i.e., ARGININE, VALINE, SERINE, and ASPARTIC ACID) is incorrect since it refers only to the L enantiomers. This should be corrected.

In the case of the L-PROLINE, the stereochemistry of the stereogenic center is not specified.

In the case of L-LYSINE and L-THREONINE, the hydrogen atom at the stereogenic center has been drawn, differently than in the case of the other amino acids. A similar representation should be used for all the amino acids.

In the case of the L- THREONINE, the stereochemistry of the stereogenic carbon bearing the hydroxy group has not been defined.

In several figures in the main text, H2SO4 should be written subscripting the numbers 2 and 4 in the chemical formula.

In the Abstract section, the authors say “…the anionic 5,10,15,20-tetrakis(4-sulfonato-phenyl)porphyrin (TPPS4) into nano-tubular J-aggregates under strong acidic condition…” but I guess that it would be more correct to remove the adjective “anionic” since at the pH at which the experiments have been carried out, the porphyrin is not in its anionic form. Actually, would it be more precise to write it as H4TPPS4?

In the Abstract section, the authors say “…with a positive chargeable side group…”. I guess that “…with a positively charged side group…” would be better.

In the manuscript, the authors say “…, Arg, Lys and Orn have all their functional groups in a protonated form, therefore they are dicationic species.”. I would suggest rephrasing this sentence. Saying that these three amino acids have all their functional groups in a protonated form is misleading (also the carboxylic acid is a functional group, but clearly it is not protonated).  

Throughout the manuscript, the word “aminoacids” is used, although I guess that the correct one (accepted by the IUPAC) is “amino acids”.

Author Response

We would thank both the Reviewers for their comments on the manuscript and suggestions for revision, that prompted us to improve it, answering to their points.

Point 1: The references section needs to be thoroughly revised. In some cases, the DOI is reported, in other cases not. In some cases, the name of the journal is not included (e.g., references 60, 61, 62, 67, etc.). Some references are duplicated (I spotted references 54 and 55, and references 30 and 40). In some cases, the journal names are abbreviated, in many other cases not (e.g., references 72, 76, 71, 63, etc.).  In some cases, the titles appear with only the first letter capitalized, in other cases the first letter of every name is capitalized. The authors are invited to revise all the references and change them accordingly following the International Journal of Molecular Science references guidelines.

Response 1: The references have been modified according to the reviewer comments and following the International Journal of Molecular Science guidelines.

Point 2: In Table SI1, the molecular structure of the amino acids with D and L handedness (i.e., ARGININE, VALINE, SERINE, and ASPARTIC ACID) is incorrect since it refers only to the L enantiomers. This should be corrected. In the case of the L-PROLINE, the stereochemistry of the stereogenic center is not specified. In the case of L-LYSINE and L-THREONINE, the hydrogen atom at the stereogenic center has been drawn, differently than in the case of the other amino acids. A similar representation should be used for all the amino acids. In the case of the L- THREONINE, the stereochemistry of the stereogenic carbon bearing the hydroxy group has not been defined.

Response 2: Table SI1 has been changed according to the reviewer comments.

Point 3: In several figures in the main text, H2SO4 should be written subscripting the numbers 2 and 4 in the chemical formula.

Response 3: H2SO4 have been written subscripting the numbers 2 and 4 in the chemical formula in figures and in the main text.

Point 4: In the Abstract section, the authors say “…the anionic 5,10,15,20-tetrakis(4-sulfonato-phenyl)porphyrin (TPPS4) into nano-tubular J-aggregates under strong acidic condition…” but I guess that it would be more correct to remove the adjective “anionic” since at the pH at which the experiments have been carried out, the porphyrin is not in its anionic form. Actually, would it be more precise to write it as H4TPPS4?

In the Abstract section, the authors say “…with a positive chargeable side group…”. I guess that “…with a positively charged side group…” would be better.

Response 4: In the Abstract section the two sentence have been modified according to the reviewer suggestions.

Point 5: In the manuscript, the authors say “…, Arg, Lys and Orn have all their functional groups in a protonated form, therefore they are dicationic species.”. I would suggest rephrasing this sentence. Saying that these three amino acids have all their functional groups in a protonated form is misleading (also the carboxylic acid is a functional group, but clearly it is not protonated). 

Response 5: The sentence “…, Arg, Lys and Orn have all their functional groups in a protonated form, therefore...” has been changed with “…, Arg, Lys and Orn have all their amino groups in a protonated form, therefore...”

Point 6: Throughout the manuscript, the word “aminoacids” is used, although I guess that the correct one (accepted by the IUPAC) is “amino acids”.

Response 6: “aminoacids” has been changed with “amino acids”.

Reviewer 2 Report

 In this work, the authors reveal that the chiral induction by aminoacids in porphyrin j-aggregates, which is an interesting topic. Some problems need to be considered before publication.

(1)   The changes in pH values can lead to a J-aggregates. What is the physical mechanism?

(2)   What are the reason for the change in extinction spectra? To better compare this point, their relative intensity can be listed as a comparison.    

(3)   Different form in Fig. 1(b), some slight change in Fig. 2(b) can be observed. What is the reason?

(4)   How to define the change in symmetry change in your work? Some additional descriptions should be provided.  

Author Response

We thank this Reviewer for his/her comments and suggestions, that give us the opportunity to explicit better the implications of our work.

Point 1: The changes in pH values can lead to a J-aggregates. What is the physical mechanism?

Response 1: The formation of J-aggregates of this particular porphyrin (TPPS4) has been extensively investigated and the aggregation mechanism is well established in literature by contribution from our group and others (see e.g. ref. 12, 14, 15, 17, 20, 21, 22, 23, 24, 28, 52 and 53). Protonation of the porphyrin core leads to the formation of a zwitter-ion that is able to interact through electrostatic interaction with other units, growing through dimerization and subsequent oligomerization that are the rate-determining steps to an auto-catalytic process leading to the eventual nano-assemblies (see Scheme 1). These are further stabilized by hydrogen-bonding and solvophobic interactions. In the main text, on p. 1-2, l. 38-48 and then on p. 2 l.84-93, at the beginning of the results and discussion section, a detailed description of the mechanism have been reported.

Point 2:  What are the reason for the change in extinction spectra? To better compare this point, their relative intensity can be listed as a comparison.

Response 2: The changes reported in Figure 1a and 2a are relative to typical extinction spectral changes occurring during the aggregation of the TPPS4 porphyrin with L-ser and L-Arg, as representative examples of two different kind of aminoacids, having different behavior. As outlined in the main text, L-Ser induces aggregation with spectral features that are typical for J-aggregates where a narrow J-band is usually found in literature for this kind of systems. L-Arg leads to a much wider and diffuse J-band, which is in line with a different electronic coupling among the units in the nanoaggregates. The differences are described in the main text (see p. 3, l. 112-116; p. 4, l. 128-137. The different values (and relative percentage changes) for the final extinction values in the various experiments are reported in Supporting Info Fig. SI3.

Point 3: Different form in Fig. 1(b), some slight change in Fig. 2(b) can be observed. What is the reason?

Response 3: The intensity and shape of the induced CD band observed for the J-aggregates in the presence of the two different typology of aminoacids reflect the different kind of electronic coupling among the chromophores in the nanoaggregates and the differences in chirality transfer and expression. The nature of these interactions has been extensively discussed in the main text (see. P. 3 l. 124-127 and p.4 l. 138-152).

Point 4: How to define the change in symmetry change in your work? Some additional descriptions should be provided.  

Response 4: This work is dealing with the expression, transfer and amplification of chirality from a templating chiral species (the aminoacid) and a growing supramolecular assembly. We suppose that the reviewer is referring to the chirality of the nanoassemblies and this is clearly indicated by the signs of the induced CD bands (that could indicate P or M helicity). This point is extensively discussed in the main text on p. 2, l. 53-81.